# Always Pay Attention to Which Model of Motor Learning You Are Using

**DOI:** 10.3390/ijerph19020711

**Published:** 2022-01-09

**Authors:** Wolfgang I. Schöllhorn, Nikolas Rizzi, Agnė Slapšinskaitė-Dackevičienė, Nuno Leite

**Affiliations:** 1Department of Training and Movement Science, Institute of Sport Science, Johannes Gutenberg-University Mainz, 55099 Mainz, Germany; nirizzi@uni-mainz.de; 2Department of Sports Medicine, Faculty of Nursing, Medical Academy, Lithuanian University of Health Sciences, Tilžės g. 18, 47181 Kaunas, Lithuania; agne.slapsinskaite-dackeviciene@lsmu.lt; 3Reseach Center in Sports Sciences, Health Sciences and Human Development (CIDESD), Department of Sport Sciences, Exercise and Health, University of Trás-os-Montes and Alto Douro, 5001-801 Vila Real, Portugal; nleite@utad.pt

**Keywords:** motor learning, repetitive learning, discovery learning, methodical series of exercise, methodical game series, variability of practice, contextual interference, differential learning, pragmatism, replication crisis

## Abstract

This critical review considers the epistemological and historical background of the theoretical construct of motor learning for a more differentiated understanding. More than simply reflecting critically on the models that are used to solve problems—whether they are applied in therapy, physical education, or training practice—this review seeks to respond constructively to the recent discussion caused by the replication crisis in life sciences. To this end, an in-depth review of contemporary motor learning approaches is provided, with a pragmatism-oriented clarification of the researcher’s intentions on fundamentals (what?), subjects (for whom?), time intervals (when?), and purpose (for what?). The complexity in which the processes of movement acquisition, learning, and refinement take place removes their predictable and linear character and therefore, from an applied point of view, invites a great deal of caution when trying to make generalization claims. Particularly when we attempt to understand and study these phenomena in unpredictable and dynamic contexts, it is recommended that scientists and practitioners seek to better understand the central role that the individual and their situatedness plays in the system. In this way, we will be closer to making a meaningful and authentic contribution to the advancement of knowledge, and not merely for the sake of renaming inventions.

## 1. Introduction

A recurring and often heated discussion among sports scientists, physical education teachers, coaches, and therapists addresses the question of the transferability of research findings to teaching students, coaching high-performance athletes, or treating patients [1]. Debate continues as to whether—and which—motor learning models developed in the laboratory can be applied in certain everyday situations [2,3]. Fundamental differences can be observed between English-speaking and European mainland countries, where for historical reasons pedagogy and sport pedagogy are much more integrated into teaching physical education in school [4].

Epistemologically, the question of transferability corresponds to a fundamental philosophical problem, namely the extent to which knowledge gained may be generalized. Thereby, the process of mapping an original to a model and the simplification carried out in the process basis play a special role. Unfortunately, a clear separation between the original and the always-simplified model is still mostly missing, neglecting the subjective part of the modeler and the epistemological basis. More recently, a prerequisite or weaker form of generalization has come under extensive discussion with the emergence of the replication crisis [5,6]. First made public in medicine [7] and psychology [8], the discussion describes a methodological crisis that has now spread across several scientific fields. This crisis stems from the revelation that only a minority of studies can be replicated, thus violating an essential requirement for research, especially in the sciences that orient themselves on classical physics and that exhibit some physics envy [9,10], although the characteristics of their research objects differ substantially [11]. The term “Envy of physics” relates to the envy by scientists in other disciplines of the mathematical precision of fundamental concepts achieved by physicists. It is a criticism leveled against academic areas, such as social and life sciences, that attempt to express their fundamental concepts in terms of mathematics, which is seen as an unwarranted push toward reductionism [12].

The low replication rates of about 11–45% [13] in the life sciences triggered a discussion [7,14] that also brought replication rates in sports science into view [15,16]. However, while in the other life sciences, various causes for the crisis have been discussed and alternatives have been proposed which have the potential to bring about change and progress, most sports science publications prefer to persevere with traditions. The majority of sports science publications still blindly follow “the ritual of mindless statistics” [17,18,19,20,21]; or propose to reintroduce long-lost faith in the Laplace demon that relies on the belief in predicting the future by knowing all laws of nature and all initial conditions, by collecting as many boundary conditions as possible, that have been and are being used in intervention studies to make motor learning predictive [22,23,24,25]; or seek to renew the belief in absolute falsification according to Popper’s theory of absolute truth [26,27] despite the fact that the corresponding positivism and neo-positivism was already overcome in the middle of the 20th century and led to post-analytic philosophy.

The most prevalent reasons given for the replication crisis outside of sports science can be broadly assigned to three categories. The somewhat conservative category cites deficient implementation of available statistical methods [17,28,29] and proposes stronger use of these methods, combined with stricter standards in the form of lower thresholds for statistical decisions [30], to improve the probability of positive replication through stronger selection. The more tolerant category calls for reducing publication bias and the related “file drawer problem” by publishing more replication studies and studies that are more likely to confirm null hypotheses, to put the statements of the significant studies more into perspective [7,31,32]. The somewhat revolutionary category questions replication as a general criterion for all sciences, with a clear reference to time scales and the closely related properties of life, and calls for spatially and temporally appropriate replication criteria [11,13,33,34,35]. This approach is advocated primarily by researchers in the life sciences whose objects of study are subject to cultural differences or constant change, such as on learning, ontogenetic, or phylogenetic time scales. 

Although the ideas of the first two categories tend to adhere to the neo-positivistic idea of gaining knowledge exclusively by falsification and have been discussed at length and urged without great success, the third (i.e., “revolutionary”) category offers the potential for change due to its consideration of an aspect that has been largely neglected so far. This is primarily due to the differences in the research objects of physics and chemistry on the one hand, and of the life sciences on the other [33]. The difference is of particular interest because the predominant discussion of epistemological issues has long been oriented toward the characteristics of knowledge discovery in mechanistic, reductionist physics and chemistry. The research objects studied in the life sciences bear fundamentally different characteristics of temporal change from those in physics and chemistry, and thus require a different replication criterion. This applies in particular to studies of motor learning. Although the fall velocity of a rigid body can be replicated under the same conditions without problems within the scope of instrumental measurement accuracy, replicating the learning of a movement is difficult due to biological changes in the body as well as memory. Although the repeated learning of a movement by the same person would likely most closely represent direct replication, the initial conditions change fundamentally with the first learning. Despite the constant changes of living systems being known, the extent of such changes has so far received little consideration in the discussion of replication. This concerns in particular the aspect of the exchange of information in communication as it occurs in a learning process [36]. The majority of studies on motor learning focus only on objective information from the external point of view and widely neglect the constantly changing subjective information of the learner [37,38].

All arguments considered together neatly mirror and confirm the pragmatic argument against immutable principles [39] and the Duhem–Quine Thesis, which, with the holistic argument considers verification or falsification by single experiments or empirical observations to be impossible [40,41,42]. While pragmatism invalidates the absoluteness of cognition by introducing an interpretant, besides signifier and sign [43], Duhem and Quine see the problem rather in the impossibility that an observation or an experiment based on a model could completely capture all influencing variables of the original within the framework. Thus, Newton’s law of gravity developed on rigid bodies is no more disproved by a falling-down than repetition learning is disproved by interleaved learning. As in the first case, the results of the second are tied to the properties of the object of study and the conditions of study. If, in one case, it is the ratio of the surface to the weight and the surrounding properties of the medium, in the second case it is the properties of the living being such as age, learning experiences, degree of openness to new things, need for security, complexity of the movement to be learned, and much more on which the result will depend. In both cases, the findings interpreted by creative scientists lead to extensions of existing knowledge—rather than to its refutation—by adding another variable and thereby modifying the model. Interestingly, this discussion has already occurred in the field of philosophy about a century ago that was strongly influenced by pragmatism and neo-pragmatism, whose founder was Charles Sanders Peirce (1839–1914).

The effects of Peirce’s [43] introduction of an interpretant in movement research will be elaborated and discussed in the following article with respect to two aspects for selected motor learning approaches: the investigator and the object of research (i.e., the person under investigation). The subsequent deliberations are based on Stachowiak’s general model theory (GeMoT) [44], which was proposed in the 1970s in response to the discourse on Popper’s falsification and the underdetermination theorem (Duhem–Quine). By considering more extensive conditions of the research context, this approach defines the locality of the expected knowledge and prevents inadmissible generalizations. Therefore, the aims of this critical review are: 

(1) To introduce two areas of research whose subject matter is closely related to the implications of Peirce’s “interpretant” for theories of motor learning and their implementation. Both provide the criteria for a template against which selected motor learning theories are discussed:(a)To derive criteria from the GeMoT for a clearer restriction of the previous generalizations of motor learning models related to the researcher as “interpretant”.(b)To derive criteria from Cybernetic Pedagogy for differentiating objective and subjective information in the context of a motor learning process related to the learner as “interpretant”;

(2) To illustrate and discussing the implications of the importance of these two areas on the most common motor learning and teaching approaches that:(a)are found in textbooks on physical education, the training of athletes, physical therapy, and occupational therapy,(b)introduce new elements related to the physical exercise process, and(c)are subjects of scientific research;

(3) To suggest an alternative approach to problems particularly related to the replication crisis through recent developments in the recognition of motion patterns using artificial intelligence (AI). This approach offers an alternative in dealing quantitatively with the object of research of moving and learning humans in the form of locally generalizable statements related to the non-repeatability of events, which are commonly and somewhat succinctly attributed to the factor of time. 

## 2. Methods

### 2.1. Objective and Theories for Criteria Templates

The objective of this study is a critical review [45] on the most common motor learning approaches based on criteria that have only sporadically been considered in the previous research. These criteria are derived from two theories that can be traced back to the theory of semiotics according to Peirce [43], namely the GeMoT and the Cybernetic Pedagogy. 

Due to the breadth of the topic and the limitation of space and time within every review, the individual areas can only be touched upon. Naturally, the older motor learning theories have a more extensive historical context, whereas with increasing topicality the scientific foundation widens in scope. This work makes no claim to completeness.

#### 2.1.1. The General Model Theory 

The GeMoT was developed by Stachowiak [44] in response to the substantial critique of several inconsistencies associated with the assumptions and coherence of analytical philosophy as represented by Popper, proponents of the Vienna circle, critical rationalists, logical empiricists, and positivists [46,47]. In essence, the fundamental critiques raised by Duhem, Quine, Kuhn, Lakatos, and Feyerabend [48], which led to post-analytic philosophy, were reaffirmed and, under the influence of cybernetics and pragmatism, combined with the development of a more comprehensive understanding of the process of modeling. As well as Duhem and Quine’s holistic argument against the possibility of falsification in principle, Peirce [43] anticipated many modern scientific and philosophical developments with his foundation of semiotics and pragmatism [46]. The essential feature, especially in the context of motor learning in the modern view, was the introduction of an interpretant as well as symbol and sign. This introduction is important not only for the learner when interpreting stimuli against the background of their history and current activity, but also (and especially) for the researcher when designing an experiment and interpreting phenomena. 

By reflecting on the historical and sociological embedding to which every interpretant (e.g., teacher, coach, therapist, and athlete or student) is exposed, the subjectivity of all knowledge is evident. Pragmatism considers knowledge as always dependent on a situated context and only understandable in relation to the intentions, purposes, and aims of the investigators. The neo-pragmatist model of cognition according to Stachowiak [44] appealingly relativizes the mapping thought of classical epistemology in the sense of the pragmatic decision. According to this model, all cognition is cognition in and by models, and, beyond other model theories, cognition is intentionally selective relative to certain subjects and temporally limited. Empirical knowledge is always understood as “knowledge for whom” and “knowledge for what purpose” under “which historical circumstances” [44]. 

From critiques of the assumptions of analytical philosophy, Stachowiak developed one of the most elaborate classifications of the model with notable implications for Cybernetic Pedagogy. However, it was “widely ignored” [49] because the theory was never translated outside German-speaking countries [50]. Nonetheless, his theory recently began to attract international interest because of its applicability in the context of the replication crisis and the rise of multimedia learning [51]. 

#### 2.1.2. The Cybernetic Pedagogy

The application of the GeMoT and its underlying philosophy also massively influenced Cybernetic Pedagogy, which gained renewed interest with the development of Educational Cybernetics and the currently booming online learning via social media [37,38,52]. Many findings from the field remain topical, although cybernetics could not establish itself as a science. Nonetheless, with the increasing digitization of our lives, most of the previously developed principles have entered our everyday lives almost unnoticed. In parallel, learning has become increasingly personalized [53]. In Pedagogical Cybernetic theory, it is assumed that the learner will gain mastery over their behavior in specific contexts by detecting differences between their earlier and current perceptions, executions, and experiences [52]. Instead of applying first-order cybernetics as signal transmission regulated by a feedback loop with external feedback from the sender, more emphasis was given to the learner as the receiver and interpretant by shifting the focus from objective to subjective information. This has been termed the cybernetics of “observing systems,” or second-order cybernetics, as an expansion of the cybernetics of “observed systems” with the understanding of objective information [54,55] on which motor learning research stalled for a long time, and was therefore likely not pursued further. 

When interpreted generously, since it was never explicitly associated with this cybernetic point of view, subjective information was at best accounted for by differentiating findings between beginner and advanced learners, children and adults, or specialists in different areas. However, research focusing on averages essentially prevented the research focusing on the learning success of the individual, which is indispensable for pursuing the idea of subjective information. Whereas classical cybernetics, such as closed-loop learning [56] and most motor learning research, was oriented towards Shanon’s [57] objective concept of information and moved into an impasse, Cybernetic Pedagogy [36,55], in accordance with pragmatism, leans toward Wiener’s [58] subjective concept. In contrast to Shanon, who understands information as a selection process based on probabilities and equates it to entropy, Cybernetic Pedagogy relies on Wiener’s understanding of information as a process of decreasing entropy or neg-entropy [59]. Thus, Shanon interprets a finite schema as a source of information that yields more information as entropy increases, whereas Wiener interprets a finite schema as a variable state of order that contains more information as the neg-entropy process increases [38]. Neg-entropy, in turn, corresponds to redundancy and is “…directly related to that psychological quantity which is interpreted as intelligibility. This increases proportionally to the redundancy. A maximally redundant message would be fully intelligible, but at the same time banal” [60]. Accordingly, a completely identical movement repetition has maximal redundancy and, therefore, does not contain any subjective information for learning. In consequence, only when repetitions are not identical and show differences, which Bernstein [61] metaphorically described as repetition without repetition, is redundancy reduced for the learner while increasing subjective information as their complimentary so that learning can occur. Learning thus acquires a strongly subjective component and can be understood as a process of gaining redundancy that depends on the information already available and corresponds to a decrease in subjective information. The subjective information of the individual learner as the interpretant of the signs is further differentiated as the subjective repertoire, subjective conditional probabilities, subjective knowledge in advance, and the invariance of the rate of absorbing subjective information [37,38,62]. Differentiating subjective information thus anticipates many approaches that later address its aspects. These will be discussed more extensively later, along with specific motor learning theories. 

### 2.2. Procedure

Using a liberal neo-pragmatic lens, the concept of cognition is restricted to a concrete area within the framework of Stachowiak’s GeMoT [44], and its respective special occupancy is defined. This is an open system insofar as the intentions, purposes, and goals are decided within the framework of modeling in the context of a group of people with sufficiently similar intentions for a certain time, related to the model criteria. Models are not simply assigned to their originals (what?). They fulfill their substitution function:(a)for certain subjects (for whom?);(b)within certain time intervals (when?);(c)for certain mental or actual operations (what for?).

By considering the investigator’s circumstances for modeling, the context pattern under which modeling takes place is emphasized and further relativizes the possible claim to the generalizability of study results [44]. 

Although these questions are rarely addressed explicitly in the extensive literature about motor learning or physical education because finding objective knowledge and pursuing the general validity of the respective learning theory dominate research practices, the specific contextuality was indirectly derived from the first publications or reports found on the respective approaches, the research-specific information about the authors that was available, and the accompanying historical conditions. Consequently, as in the shift in learning studies under the paradigm of behaviorism from generalization [63] to discrimination learning [64], the claim for validity with this kind of modeling shifts from the general to highly specific claims that are also determined by different context patterns.

What is modeled was derived not only by the name of the given motor learning approach but also by the specifics of the first experiments or reports found (if available). Because most motor learning models of interest began with claims of generality in terms of both the object of modeling (what?) and user reference (for whom?), both criteria were derived from the descriptions of the experimental setup and the primary readership of the publication. So as not to limit the claim to validity or the degree of generalization too much, criterion categories were chosen, which also provided evidence for corroboration in several studies as influencing variables for motor learning.

The historical–social boundary conditions (when?) under which the models were developed were identified by the year of the first publication. The original intent (what for?) was derived from both the user reference and the historical–sociological boundary conditions.

For delineation within the frame of the GeMoT, the most common motor learning models used in sport pedagogy, physical education, sport psychology, and sports science were included. The inclusion criteria were that the models had to contribute innovations for their time and had to lead to obvious changes in the content or sequence of physical exercises. The models can be coarsely described as a historical collection of physical search strategies to solve specific movement problems [65,66,67] and are distinguished here from more mental approaches that focus on exclusively cognition-oriented criteria (e.g., motivation, stress, anticipation, or emotional issues in learning), knowing that these groups cannot be separated in real terms. The terms “motor learning,” “coordination learning,” and “skill acquisition” are used interchangeably. However, from the perspective of pedagogy, teaching and learning fundamentally differ from each other, not only in perspective but also in the contribution being limited exclusively to the dominant movement-related approaches that are momentarily led by sports psychologist approaches. Going back to the roots of pedagogy in Greek philosophy, learning has always been related to self-activity and discovery. However, learning psychology and, subsequently, sport psychological learning research, due to its scientific orientation in connection with the after-effects of behaviorism, increasingly restricted itself to the optimization of measurable behavior through visual, auditory etc. stimuli. Although for a long time—at least in continental European sports pedagogy—the question of “what?” was the central object of this education in school, the question of “how?” increasingly moved to the center of interest in the context of the Cold War. Both foci resonate in the discussion between motor learning researchers and physical education teachers. While sports pedagogues or physical education teachers mainly hold compulsory lessons and always pursue multiple teaching goals in parallel, with motoric goals being one aspect alongside the cognitive, social, affective, and other goals, the sports coach rather deals with volunteers who agree to the goal of motoric performance improvement. However, as performance levels rise, athletic trainers become increasingly aware of goals for the athlete that go beyond purely motor goals that are associated with psychology, sociology, nutrition, lifestyle, injury prevention etc. 

The models of interest in the following review are: (1)Repetitive Learning (RL);(2)Discovery-based Learning (DBL);(3)Methodical series of exercise (MSE);(4)Methodical game series (MGS);(5)Variability of Practice Learning (VP);(6)Contextual Interference Learning (CI);(7)Differential Learning (DL).

Against the current state of knowledge, a group of variables that influence the results of studies on motor learning was identified, although they have not been the specific subject of investigations. Some of them were not explicitly mentioned in the original studies because they were considered normal in the historical context, and others did not appear in the literature because researchers were unaware of their influence at the time the model was introduced. In most models, the initial study was conducted to confirm a theory by an example using specific movements under specific conditions. The features that best distinguished all motor learning models by their claims are: (a)Guidance (supervised/self-organized);(b)Degrees of freedom of movement (low/high);(c)Dominant sensory system (visual/kinesthetic);(d)Processing (serial/parallel);(e)Learning stadium (acquisition/stabilization/refinement).

The motor learning models will be discussed in historical order and based on these features. 

## 3. Results

To illustrate the motor learning approaches discussed below, Figure 1 shows all approaches schematically. The letters A, B, and C each represent specific movement techniques. In RL, the same movement is only imitated and repeated. In DBL, the solution must first be found from a variety of possibilities. The MSE successively assembles the movement to be learned. In the VP, a change in the size of the letter A corresponds to a parameter variation of the same invariants in repetition blocks. CI-1 allows a randomized sequence of parameter variations. CI-2 records movements with different generalized motor programs (GMPs) in randomized order. DL for learning one technique adds random perturbations to the movement being learned. DL for learning multiple techniques applies the principle analogously for all techniques. 

### 3.1. Repetitive Learning Model (RL)

#### 3.1.1. Description

The RL model aims to map changing behavior or changing thinking by repetitively imitating a role model. This model also serves for acquiring and improving quite simple movements.

#### 3.1.2. Historical Context

The origins of learning by repetition are arguably as old as humankind and as such it is difficult to attribute this approach to individual authors or eras. In the education of children (“pedagogy” from the Greek ped, “child”), learning by repetition is mentioned from ancient times, both casually and as one of various methods. Historically, in the field of pedagogy, the content, i.e., what should be taught and learned, especially in the context of political education, always received greater emphasis than how that content should be taught. Methods of teaching were rarely the subject of discussion until the 18th century. References to RL as a method in pedagogy can therefore be derived only indirectly.

In essence, repetition learning is thought to play a dominant role in two areas of training (keeping in mind that the term “training” first arose in the 19th century in the context of horse training). The first is physical training for fighting, to achieve automation and perfection in addition to habituation with military practice [68,69]. The second is in religion, where repetition occurs by reciting religious texts and rhythmic movements to reach a trance-like state via habituation. In both cases, repetition learning is originated in—and mainly applied to—the education of adults (“andragogy” from the Greek root andr-, “man,” and agein, “guiding”), associated with orientation to role models, and is closely linked to education for obedience. For the sophists, repetition was a form of gaining knowledge through “exercise” (askesis) in interaction with “natural disposition” (physis) and “instruction” (mathesis) [70]. Interestingly, in ancient Greece, the institutions in which learning took place were called “schole”, from which today’s term “school” can be derived and which originally meant “leisure, free time, rest from work, having fun.” Through the Romans, who were best known for their strong military and only later imported the Greek culture along with Greek slaves, the Latin schola moved toward speech, and recital, and became the foundation of the modern school.

RL experienced a true renaissance under the influence of the Christian church during the Middle Ages and a period of inquisition in the form of inculcating drills through the pedagogy of the Enlightenment in the wake of Voltaire and Kant. Children were to be educated as early as possible to become beings of reason and, to this end, they were to adopt the attitude of small adults as early as possible. With the German Turnbewegung (gymnastics movement) promoted by Turnvater Jahn in 1812 [71], the exercises became more varied but still had to be repeated to perfection in military drill [72]. Drill and the disciplining of the body were considered an “indispensable part of Turnen [gymnastics]” and a prerequisite for military strength [73,74]. Something similar could be observed in sports during the Third Reich and in the Eastern Bloc during the Cold War, where increased school sport lessons were intended to achieve military strength. Correspondingly, training theory originating in the former Soviet Union and former GDR is military in nature, and particularly focused on intensity and volume with a high level of external guidance [75,76,77]. With the fall of the Berlin Wall just before the 1990s, numerous coaches that were educated in this militaristic style of repetition and obedience spread their philosophy worldwide, especially in high-performance training. Overall, repetition learning is mentioned in older literature primarily in connection with military drill and education for obedience with a high level of guidance, and it was considered the solution to all learning processes without any differentiation of degrees of freedom, complexity, or the sensory systems that are stressed. 

The first approaches to this learning model in accordance with modern scientific methods began at the end of the 19th century with more detailed descriptions of learning and forgetting curves during the repetition of cognitive content [78,79] and in the context of experiments on behaviorism in the conditioning of behavior [80]. In the motor research of the 20th century, the same is then found in movements with small [81] and large degrees of freedom (sDGF/lDGF) [82,83,84]. The accompanying phenomena were described in detail and arbitrarily subdivided into two [84], three [81,82,83], or up to six phases [61]. Characteristically, the descriptions were directly mapped to prescriptions for the learning steps in a naturalistic fallacy, which is an informal logical fallacy which argues that if something is ‘natural’ it must be good [85]. The influence of guidance by manual [86], mechanical [87], verbal [88], or visual [89] aids in the context of RL has been studied in parallel. Interestingly, this form of learning, well known to educators [90,91], is currently experiencing a renaissance in competitive sport with a new name, the constraints-led approach (CLA), where material or instructional aids are also used to constrain the learner’s movements in a way that guides him or her in the direction of the coach’s learning aim [92,93]. However, while in RL errors are to be avoided and individual techniques are tolerated only at the highest level of performance as an inevitable by-product [75,94], in CLA erroneous movements are allowed but only to experience how not to do them, and individual techniques are accepted but only within the guiding constraints [93]. Because of its prevalence and lack of alternatives, RL also served as the basis for Hebb’s [95] adaptation studies, in which he observed an increased strength of connections between two neurons with an increasing number of grazing repetitions. From the perspective of cybernetic pedagogy, RL addresses only external objective information. The learner is regarded as having a tabula rasa.

#### 3.1.3. Model Decision

The RL (or habituation learning) model has been used to train a wide variety of forms of movements in different age groups and at different stages of learning, predominantly by ancient (when?) militaries and religions to educate adults, but also in everyday life in correspondingly fewer complex forms of movement (what?). No differentiation with respect to the subject (e.g., cognitive, social, emotional, or motoric skills) or predominant senso-motoric system is evident (for what?). With only sparse alternatives, the RL model has historically been the dominant model used by the powerful to educate society by habituation. RL was transferred from andragogy to the education of children and adolescents and was initially only marginally used in pedagogy. Proponents of the RL approach still claim general applicability regardless of learning levels, age, or complexity of a movement. This is experiencing continuation in cognition-oriented learning by means of feedback on result or performance, where the notion of correct repetitions is mainly accompanied only by cognitive components [56].

### 3.2. Discovery-Based Learning Model (DBL)

#### 3.2.1. Description

DBL allows students to take charge of their learning through hands-on exploration and inquiry; instead of memorization and repetition of concepts, learning through unique experiences is emphasized. In pedagogy, DBL is typically characterized by having minimal teacher guidance, solving problems with multiple solutions, minimal repetition, and memorization [96]. In psychological research, DBL is simplified to trial-and-error learning [97].

#### 3.2.2. Historical Context

The first written references to precursors to modern DBL can be found in the pedagogical literature of ancient Athens. More recent roots appear in reform or progressive pedagogy, attributed originally to Rousseau [98] who, with his mantra, “The only habit a child may adopt is that of adopting none,” set a contrast to the military style of teaching [98]. Basedow, Pestalozzi, Froebel, Gaudig, Kerschensteiner, Dewey, Freinet, Montessori, Steiner, and Neill were the most famous representatives of reform education. Under this approach, the emphasis was on learning by doing, problem-solving, and critical thinking, rather than blindly obeying. Children were no longer understood as blank sheets of paper, but rather as independent beings who develop their characteristics in a reciprocal or transactional relationship with the natural environment [99]. The proposal of learner-dependent education introduces the interaction of two systems, which, unlike RL, allows a qualitative distinction between objective and subjective information in the learner according to cybernetic pedagogy. The introduced interaction of teacher and learner allows for a scale of teacher influence, from no control to full control. Mosston [100] subdivides this continuous scale for the field of physical education into ten qualitative levels of interaction. 

In motor learning research, DBL has been studied since the 1970s [97,101]; however, these laboratory studies were performed on adults and can be considered attempts to transfer pedagogical content to andragogy. Assessments of the results were mixed, with evaluations ranging from nearly inappropriate (in the sense of not suitable for teaching) [97,102], to conditionally appropriate [101], to unconditionally appropriate [103], without addressing the studies’ specific and diverse boundary conditions. Epistemologically, most studies are unfortunately misinterpreted as falsifications of the other theories, despite the specific differences in conditions, due to the belief that a democratic majority decision will determine the successful theory. 

Offshoots of the DBL approach are dispersed under a wide variety of names [104] including discovery [105,106], experiential [107], inquiry-based [108], problem-based [109,110,111,112,113], non-linear pedagogical [114], critical thinking [115,116], and constructivist learning [117,118,119,120], thereby constituting an attractive target for another Sokal hoax [121]. Due to the increasing prevalence of system dynamics [122,123], the associated reinterpretation of movement fluctuations, the analysis procedures developed for this purpose [124] and the derived practical applications, systematic research into DBL in motor therapy [125,126], learning [127], and training [24,128,129,130] is undergoing a veritable renaissance.

#### 3.2.3. Model Decision

DBL’s origins in Greek pedagogy and Rousseau’s philosophy (when?) indicate an intent to educate children by helping them to find solutions to problems (for what?) independent of the intended subject. Most literature related to DBL is directed toward the acquisition of movement skills (what?). Stabilizing and perfecting skills is expected through the parallel acquisition of abilities that today would be called executive functions, which are developed primarily through self-activity and self-motivation [131]. Contributions to DBL in the motoric field are primarily addressed to pedagogues, who are directly concerned with teaching, and secondarily to politicians (for whom?), who are urged to make this form of learning compulsory through state school curricula.

### 3.3. Methodical Series of Exercise Model (MSE)

#### 3.3.1. Description

The MSE model was developed in German–Austrian sports pedagogy [132] and describes approaching a movement step-by-step through increasingly difficult preliminary exercises. Within the MSE model, three approaches are distinguished. First, learning aids are successively removed; second, there is a gradual approach to the target exercise; and third, the target movement is broken down into functional subunits and successively assembled [133].

#### 3.3.2. Historical Context

The origins of this approach lie in the reform pedagogy mentioned above, which was applied to more complex gymnastic movements that could not simply be imitated and repeated as a whole. A synthesis of reform pedagogy with the various contemporary currents of physical exercise (German gymnastics, Anglo-Saxon sport, play movement, and the Swedish gymnastics movement) formed the “Natürliches Turnen,” or natural gymnastics, [132,134,135] and represents the basis of the MSE model as well as the methodical game series model (of which more later). By introducing learners to successively more complex forms of movement and play, everyone’s respective starting abilities are considered. By considering learners’ initial level of ability, experience in the form of subjective information is systematically taken into account for the first time from a cybernetic point of view. Accompanying the call for self-activity and self-development, the learning child is expected to “make erroneous movements in order to feel the correctness of the purposeful ones” [132]. By keeping the nature and number of errors indeterminate, Plato’s binary contrast learning is significantly extended, but a “correct” solution that implies an “error-ridden” movement is still assumed. From the point of view of Cybernetic Pedagogy, adjusting the difficulty level of the exercises to the learner’s knowledge or skill level at the beginning corresponds to taking subjective information into account. This is then binary scaled by considering the exercise failed or passed. External objective information still dominates the learning.

#### 3.3.3. Model Decision

The MSE model maps the motor DBL of reform pedagogy to a more teacher-directed form of successive approaches to learn a more complex movement (what?). The literature on natural gymnastics and MSE in the period between the two world wars (when?) and afterward is distinctly normative (with little ‘scientific’ but ample practical evidence) and directed exclusively at physical education teachers and educators according (for whom?) to then-standard principles in pedagogy such as “from simple to complex” and “from easy to difficult” (for what?). The difficulty of the exercises is thus adapted to the initial abilities of each learner. No differentiation is made according to the use of different sensory systems and there is no mention of exercises to use once the intended movement has been achieved. Typically, the RL model follows successful learning by MSE.

### 3.4. Methodical Game Series Model (MGS)

#### 3.4.1. Description

The MGS model (later renamed the game-based approach), similarly to the MSE, attempts to gradually address the more complex movements used in sports games by starting with games of reduced difficulty and more simplicity by incorporating a smaller number of players, smaller playing fields, or simplified rules. These are successively increased in number and/or size, and/or the number or complexity of the rules are increased following the pedagogical principle of moving from simple to complex or from easy to hard. This approach has a specific role within the canon of motor learning approaches because the movement techniques that are necessary for each game are not specifically taught; rather, they are learned in the context of a game alongside teammates, which immediately emphasizes not only visual and acoustic perception but also an increased internal load through additional social pressure and expectations.

#### 3.4.2. Historical Context

The use of games as pedagogical tools can be traced back to antiquity. In more modern times, Schiller (1759–1805) believed “that man can only be shaped into a real person through play” [136]. The reform pedagogue Froebel (1782–1852) saw in children’s play not playfulness, but high seriousness with deep meaning. Purposeful motoric learning games for the recreation of the body and mind already existed among philanthropists [137].

Within most of the proposed definitions of games [138,139], the distinction between purpose-free and purpose-directed games is of interest in this discussion [140]. Functional games in which one’s possible actions can be tested through the interaction of one’s body with the environment are considered to be purpose-free. Purpose-directed games are those that serve a learning goal but are still playful. The sports game here occupies a special position, as it can be work and a source of income on one hand, and is also associated with playfulness on the other. In the pedagogical field, play has represented a learning model of its own from time immemorial.

With the increasing spread of Anglo-American game culture in Europe after World War II, the idea of the MSE was transferred to the teaching of sports games. To provide an alternative teaching method to the predominant technique-based approach, the MGS model teaches the central ideas of sports games through a variety of smaller variations (such as small-sided games) from the beginning. With partner, group, and team games on playing fields of different shapes and sizes, as well as variable rules, the game ideas can be taught in a simplified form with increasing complexity in the spirit of reform pedagogy [141,142]. Sections for specific technology learning were included in addition to the above three principles of MSE. Under the influence of the philosophy of the Frankfurt School and its critical theory, MGS even developed into the physical education concept of open sports tuition, or physical education [143], characterized by students’ self-determination and co-determination in the selection of content, work materials, and methods. By expanding the possibilities of movements and reducing the rules within specific games compared to the previous approaches, the MGS model gives a much stronger, although still qualitative, role to subjective information in the sense of cybernetic pedagogy. 

First developed in Germany [144] and France [145], the MGS spread to English-speaking countries with Bunker and Thorpe’s 1982 publication [146]. Under the parallel influence of the cognitive turn [147,148], the MGS expanded to include interrupted games with interactive reflection periods, and became known as teaching games for understanding (TGfU). Since the first publication of TGfU literature, there have been many further developments and variations of the approach with analogous ideas, thereby stifling new thinking [149,150,151].

#### 3.4.3. Model Decision

With its origins after World War II (when?), the MGS model mapped the acquisition of gross-motor movement forms that would enable mastery of any type of team play in sports (what?). The learning of the movements necessary for this was holistic, undifferentiated by the different sensory systems, integrated into the approach from the beginning, and adapted to the different ability levels of novices through an appropriate modification of the rules (for what?). The approach was aimed exclusively at physical education teachers in schools—and thus at young people (for whom?). Through the increasingly parallel activity of physical education teachers in clubs, the approach was also increasingly applied to andragogy. The first publications were mainly qualitative-normative, which were later accompanied by quantitative descriptions of the realization with a finer scale for evaluating the previously given norms. [152]. Similar to MSE, when the target game is reached in field size and number of players, RL comes in to play and variations in tactics become the main alternation.

### 3.5. Variability of Practice Model (VP)

#### 3.5.1. Description

The VP model recommends stabilizing automatized movements by repeating invariant (INV) elements of those movements in combination with variable (VAR) parameters [153] in blocked order. The VP model relies on the schema model for motor control [154], which defines common elements of movement classes as invariants and specifies their concrete realization by means of variable parameters. In addition, two virtual models of memory were introduced, in which the recall-memory is responsible for the rules of applying the invariants and variable parameters in a specific realization, and the recognition-memory is responsible for the expected sensations. A physiological interpretation of the model led to the impulse-timing hypothesis [155,156] where the invariants, often interpreted as GMPs [157], were assigned to relative forces and relative timing as well as the sequence of muscle activation within different muscle groups. The variable parameters were defined by the absolute forces and absolute timing of muscle activation, as well as the selection of muscle groups.

#### 3.5.2. Historical Context

Historically, schema theory was developed at a time when life and work were still imprinted with the aftermath of experimental behaviorism and mixed with US Cold War pragmatism. Work and life were subordinated to the premise of effective production to demonstrate the superiority of the system. In addition to assembly line and desk work, this referred to the learning of languages in foreign colonies. Research was conducted to make them more effective and, for this purpose, standardized statistical procedures to increase research output were agreed upon, mainly by neglecting the associated epistemological foundations [29]. Under the strong influence of laboratory work during the ascendancy of behaviorism, learning was reduced to observable behavior and, thus, teaching and learning were virtually equated [158]. Single-case studies, common in much of psychology until World War II, were banished, and experiments with large samples were defined as the standard. With the “cognitive turn” of the early 1960s, the central black box for movement control of behaviorism was slowly filled with various forms of memory, but the basic idea that internal responses occur only via external stimuli as objective information still dominated. Correspondingly, much psychomotor research, even in sports science, focused on activities under similar conditions, namely at desks and mostly in the form of hand-eye coordination. Accordingly, most tasks in studies on the VP model consist of pressing buttons arranged horizontally on a table at short distances and within the reach of seated subjects [159]. The dominance of objective information by means of external stimuli is best shown by the research on learning for results, where augmented feedback is given as objective information; subjective information is neglected. However, with the advent of the CI model and its unification, research on VP became less common.

#### 3.5.3. Model Decision

The original work was published in the 1970s (when?) in a journal that was read mostly by movement researchers with psychological backgrounds (for whom?). Most papers on schema theory and the VP model focused on automatized fine motor movements with a high visual component (what?). The stability of the movements should be increased by repeating the invariants under variable conditions (for what?). Typically, young adults 20 to 30 years old were the subjects. The information for the invariants and variable parameters, although attributed to internal memory models, did not include subjective information in the sense of Cybernetic Pedagogy; rather, it was limited to externally observable objective information.

### 3.6. Contextual Interference Model (CI)

#### 3.6.1. Description

The original CI model projects the phenomenon of learning a single movement (the “text”) by means of repetitions that are interleaved randomly with other concrete movements (the “con-text”). A short-term disadvantage (“interference”) after skill acquisition is overcompensated with long-term advantages in the subsequent learning phase [160]. The former is explained by an overloaded working memory (WM), whereas the latter is most often assigned to three controversially discussed models: the elaboration [161], the reconstruction [162], and the forgetting hypothesis [163]. Meanwhile, the CI model is most often applied to the parallel learning of multiple movements whose parallel acquisition in random order leads to advantages over learning them in blocked order.

#### 3.6.2. Historical Context

Originally developed in psychological research on learning and memorizing letter sequences [160] and later transferred to the motor domain [164], the historical context was the same as for VP learning, which was developed under the legacy of behaviorism, the effort to demonstrate cognitive involvement, and the Cold War. To exclude individual pre-knowledge and the experiences of the subjects participating in the experiments, meaningless sequences of letters were used, thereby excluding subjective information. Similarly, as in most of the experiments in VP studies, either wooden blocks aligned on a table were to be knocked over, or buttons or keys had to be pressed in a given order and rhythm [159,164]. Both types of movements were executed mainly in the horizontal plane, simply with low dynamics and a large visual component, and unfamiliar to the subjects to exclude subjective information. In accordance with the RL model, successful learning in the CI model is based on the number of movements performed correctly [161]. With the incorporation of the VP model, low (blocked, or CI-1) and high (random or serial, or CI-2) CI variants of learning were distinguished. In the first form, the intended movement is interleaved by movements within the same GMP, and in the second form, between different GMPs, with greater effects attributed to the second [159]. Whereas the VP model filled the black box for memory information to eventually control movements in connection with long-term memory, the CI model instead directed its focus to how much objective information during pre- and postprocessing influences learning within a very general WM model. This relied on a WM model that was originally developed for the memorization of letter sequences and was inspired by early computer designs [165]. Consideration of the influence of activities that immediately precede or follow the movement being learned was strongly influenced by research on pre- and retroactive interference that had been current up to that time [166,167,168] and underscores the original intent to model the contextual conditions for learning a single movement. Specifications of the originally general claims were found quite early in terms of personal characteristics [169,170,171], task type [172,173,174], and experimental setup [162]. After this period of initial searching for structural variables of contextual conditions, research shifted more towards applied studies that focused almost exclusively on age and complexity of movement and widely ignored the previously found restrictions. Most frequently, three age groups (children, adults, and the elderly) were studied, and an increase in complexity was typically associated with an increase in sequential steps [175] or an increase in possible and similar combinations for decisions mainly before the execution of a rather simple movement [176]. Meta-analyses show systematic effects only for movements with sDGF and for adults [177,178,179]. Supporting evidence for both CI phenomena is provided from fMRI studies also limited to movements with sDGF and a dominant visual component [180] (for a summary, see [180]). Similar features are found in positive findings on movements with lDGF, in all of which a dominant visual component was involved, such as in badminton [181] (target focusing), golf putting [182] (visually estimating distance), or shooting [183] (target focusing).

In a remarkable step beyond the original, exclusively teacher-oriented, and objective information approaches, a more learner-oriented approach considering subjective information as influential was suggested. Again, with different terms in different research areas, the same pedagogical principle of guiding from simple to complex was applied. With the hypothesis of a moderate level of risk for high achievers [184], desirable difficulties [185,186,187,188], or the challenge point [189], the individual conditions of the learner were incorporated, albeit mainly metaphorically. In all cases, the basic idea of reform pedagogues was adopted similarly to the MSE model and realized by an increasing difficulty that is adapted to the individual’s abilities. This, in turn, corresponds to the general principle of Cybernetic Pedagogy, according to which the subjective information to be assimilated per unit of time is an individual constant [37,38,62,190]. The subjective information absorbed in a first learning step represents an evolved redundancy for the learner. In order to achieve the most effective learning process for the individual, the amount of information for the next task would have to be increased by exactly this amount in a further learning step, which means it would have to contain something subjectively new. Structurally, the need for novelty and subjectivity for successful learning is thereby acknowledged, but the CI model is silent on how this is met after the random order stage has been reached. From a biological point of view, this differentiation in comparison to acquisition or refinement is expressed via accommodation, in contrast to assimilation, wherein new structures must be found [105]. Regardless of these limitations, the two phenomena of intra-task interference and inter-task facilitation associated with the CI model inspired the entire research field with the standardized distinction of acquisition and learning phases in most motor learning studies [191].

#### 3.6.3. Model Decision

The first series of publications at the beginning of the 1980s (when?) was published exclusively in scientific books and journals for movement researchers with psychological backgrounds (for whom?). Similar to the original learning of letter sequences, learning in the CI model was exclusively understood as a specific memorizing process by means of correct repetitions of movements with sDGF and the high involvement of visual aspects (for what?). To date, the acquisition of a completely new skill that had not been mastered before—such as learning to ride a bicycle or a Tsukahara vault in artistic gymnastics—or how to modify a stereotyped movement for improvement, has rarely been the subject of CI research [192] and is hardly included by the CI model. Most initial experiments were conducted with adult subjects. The systematic corroboration of the CI model was achieved by sequential movements with sDGF and a dominant visual component (what?).

### 3.7. Differential Learning Model (DL)

#### 3.7.1. Description

The DL model is based on the idea that learning requires differences that are facilitated by adding stochastic perturbations during the learning process [130,193,194]. Dependent on the individual’s situation, the DL model in its most extreme version is associated with no repetition and no augmented feedback, to allow a real self-organizing process where no explicit guidance by an external agent about the solution is given to the athlete or learner, which includes not even by indicating errors [195]. Shortly thereafter, the initial mere amplification of noise is differentiated into the mutual optimal tuning of two noisy signals, one coming from the instructed or chosen exercises that correspond to the objective information, the other caused by the learner’s movements as subjective information [194,196,197,198]. The tuning could potentially be described by the model of stochastic resonance [199,200].

#### 3.7.2. Historical Context

The following explications are somewhat longer than the previous ones because the origins of the DL model derive from a variety of other research areas, the majority of which are themselves of recent origin, and therefore it is not feasible to draw on previous discussions.

Historically, a re-interpretation of Bernstein’s [61] mantra “repetition without repetition” led to the reimagining of data on continuous variance in movement [201,202,203,204,205]. Through the end of the last century, greater tolerance developed for movement variability; the previously detrimentally interpreted variances observed in various life sciences [206] were accepted and renamed “functional variability” [207] or “essential noise” [208]. The stagnation of this descriptive process resulted mainly from two problems. The first problem lay in the difficulty of transferring the evolving system dynamic approach advocated by the research groups around Kugler, Haken, Kelso, and Turvey for cyclic movements with sDGF to ballistic movements with lDGF. The second problem stemmed from unifying the principles of system dynamics, which are largely based on the constructive role of fluctuations, with models wherein these were understood as errors that needed to be minimized, e.g., by attempting to incorporate the CI approach [93,205,209] or for control problems on goal-directed arm movements [204]. A consistent integration of the CI model into system dynamics, among numerous other inconsistencies, is yet to be achieved within the eclectic ecological dynamics and CLA approach [93,210].

In parallel, biomechanical analyses became increasingly affordable and standard in high-performance sports and the first AI methods (e.g., MLPs and ANNs) were applied to the analysis of ballistic and cyclic whole-body movements [196,211,212,213]. Whereas the former provided the basis for the necessary amount of data that enabled the application of increasingly sophisticated tools to describe fluctuations with nonlinear models [214,215], the latter opened a new door for analyzing large data sets, as they accrued in series of gross motor studies and had become standard tools in the meantime [216]. 

Out of this framework, the DL model was derived from the logical contradictions of biomechanical analyses of high-performance athletes [217,218,219] by means of ANNs [211,219,220,221]. The identification of world-class athletes based on their movement patterns [211,219], without finding two identical movements, but instead through their individual and situational dependence on emotions [222], fatigue [223,224], or timing [225,226,227], deconstructed [228] the two major assumptions of the classical theory of training and therapy originating in the Stalinist-era educational philosophy of the former Soviet Union [75]. As a further consequence, even the probability of finding timeless and person-independent key variables [23,210] became nearly impossible [229]. A solution for these contradictions was suspected by applying elements of the theory of system dynamics [122]. In consequence, the most identical possible repetitions of supposedly correct movements were replaced by increased fluctuations, and the ideal to be imitated was replaced by a constantly changing individual movement pattern, which emerges through a self-organization process. Thereby, the fluctuations were amplified by adding stochastic perturbations. A stochastic perturbation is understood here as any random sequence of variations in elements of movement, where the perturbations may have an internal or external origin. In this context, the term random is interpreted in a mathematical sense, which also includes repetitions of numbers, as well as strictly monotonic or cyclic sequences that are not considered as such in everyday life [197].

The first pilot studies of the DL model were conducted on single techniques in sprint running [230], shot put [231], basketball [232], volleyball [233], and football [234]. Although the first two studies each led the athletes to performance ranges that had not previously been achieved and forced them to assimilate or refine their techniques for higher running speeds and greater shot distances, the latter three studies revealed an accommodation through a reduction in target dispersion, or stabilization, through greater precision in passing or kicking the ball towards a target. Most intriguingly, both types of learning, stabilization, and acquisition/refinement, were achieved without having executed a single presumably correct, movement and therefore massively contradicted all previous repetition-based models. Recently, two studies provided evidence for the advantageous effects of DL on brain activation in comparison to repetitive, blocked, or random CI [235,236] and may show parallels with body–mind exercises from the Far East [237].

Meanwhile, increased noise as an active intervention tool in training and therapy has been applied successfully with different names in fine motor movements [238] (structural learning), children’s attentional focus [239] (life kinetic), football [240] (dynamic brain/creativity training), team sports [114] (non-linear pedagogy), and clinical treatment [241,242,243] (perturbation training). Also meanwhile, early skeptics became proponents of the benefit of external noise on motor learning [244,245] and provided further corroboration of the system dynamic idea of the constructive influence of interacting with internally and externally caused fluctuations on learning [130,193]. Other studies [127,246] exclusively investigated the influence of a learner’s individual noise on their subsequent learning of target movements with sDGF.

#### 3.7.3. Model Decision

The DL model was published in the late 1990s (when?) in high-performance sports journals read mainly by high-performance coaches (for whom?) [130,193]. The model was presented simultaneously at biomechanics [128] and motor control conferences [247] for scientific discussion. The first experiments concerned stabilizing and refining sports techniques (what?). Although the original DL model was mainly published in journals for high-performance coaches, due to its applicability to quite universally valid physical, chemical, cybernetic, and neurophysiological theories, a more general claim was made towards learning in general (for what?). Despite the strategy of structural replication studies based on the plurality of reasoning strategies [248,249,250] that are accompanied by greater heterogeneity than direct replication studies, a recent exploratory review that expects homogenous studies [251] provided further evidence for corroboration of the DL model. Thus, the comparison of the DL model with some later derived approaches that represent only subsets of the original model did not reveal any significant differences.

### 3.8. Graphical and Tabular Summary of the Results

According to the model decisions of the scope of all discussed motor learning approaches, Table 1 and the scheme of Figure 2 result for the different time scales of learning. It is important to note here that a distinction is made between two timescales, while time scale 1 is in the range of learning units and learning segments over days and weeks, time scale 2 refers to actions that are executed immediately before or after a movement execution. In Figure 2 the pre- and retroactive model is also included to display their different time scales. If these measures are executed before and after a training session, time scale 2 merges into time scale 1.

## 4. Discussion

In the context of the previously discussed replication crisis, which is primarily affecting the life sciences, this review was intended to critically reflect on the claims for generalization of the most common motor learning models that are used in therapy, physical education, and training practice. Overcoming the epistemologically widely outdated understanding of the science of positivism and neo-positivism, the GeMoT according to Stachowiak [44] was outlined based on pragmatism and neo-pragmatism and exemplarily applied to these models. The consistent reference to the often-neglected importance of the interpretant within Peirce’s theory of semiotics led to various consequences. The broader context at the time of drafting and the original formulation of the models were largely indicative of the scope of the generalization claims. In particular, greater consideration of the intentions and motivations of researchers as constructors of the models according to Stachowiak’s GeMoT led to greater specifications of the generality of the model claims that were later supported by studies. Reflecting on the specific and structural model contents revealed extensive research potential that is mainly related to the concentration on external objective information and overall neglect of learner-related subjective information. The motor learning models discussed can be roughly classified into three groups by their original fields of interest.

### 4.1. Models Originating in Sport Pedagogy

The first group is formed by the models stemming from normative master doctrines in pedagogy and andragogy that were influenced by the respective philosophies of the ideas of man and were more focused on contents than on teaching method due to their associated socio-political intentions. Interestingly, the idea of repetition is more often associated with andragogy and obedience for socio-politically motivated aims, whereas variable, explorative learning has been part of pedagogy from early on. Historically conditioned, these models’ proofs were historically conditioned by exclusively following their prescriptions, mostly in the absence of alternatives with which to compare them. The teaching philosophy that originated in andragogy was based mainly on the metaphor of a tabula rasa lacking subjective information, which had to be written on with externally provided objective information. Failing to achieve the intended learning aims was typically ascribed to a lack of effort on the learner’s part through too few repetitions or for lack of adequate mental fortitude. The parallel achievement of multiple aims (e.g., coordinative, cognitive, affective, social, or political aims) independent of time was sufficient justification. The long history of RL and habituation is probably one reason why this model most often serves as a reference in the form of a control group in modern learning science. The same, meanwhile, applies to motor learning approaches that are based on classical sports pedagogical principles when movements are growing more complex. In modern times, reform pedagogy attributes individuality to learners and, thus, subjective information and an adaptation of objective information as well. With the spreading of the “science virus” claiming evidence-based knowledge [252], greater experimentation has been observed in physical education in school and greater pedagogization in adult’s therapy and high-performance sports, especially in English-speaking countries [4]. The first case is often a matter of confirming the effectiveness of existing models related only to the motoric aspect and accompanied by the development of more sophisticated analytical tools. In contrast, in the second case, the models serve as inspiration for a comprehensive expansion of intervention tools in these areas for a more individualized treatment that is largely independent of scientific discussions. In addition to integrating and understanding this diversity of inspiring ideas with reform pedagogy, a future challenge will be to translate the always multidimensional teaching goals in sport pedagogy—such as affective, social, political, cognitive, and motor—into models that additionally depend on various individually and constantly changing conditions.

### 4.2. Models Originating in Sport Psychology

The second group of models comprises motor learning models that, in the wake of experimental behaviorism and in the context of the Cold War, clearly shifted away from single cases of prewar psychology toward group studies. Psychomotor research focused on movements related to desk and assembly line work with sDGF and a large visual component. This group of motor learning models arose in the aftermath of the cognitive shift to emphasize the processing of cognitive information, but exclusively claimed the learning of coordinative aspects. However, with an increasing number of attempts to expand the scope of validity and thus the scope of application of these models, the original, general claims became more specific. In most cases, the application areas confirmed those included in the first published studies. According to Duhem and Quine’s holism argument, instead of falsifying the models, each study helped to increase the specificity of the original model by adding new criteria that had been neglected originally. More specifically, regardless of the original movement examples, the VP model was generalized to all movement forms [253] and their acquisition for school purposes without experimental testing [254,255,256]. Generalization was performed, resulting in neglect of, for example, the biomechanics of forces that cause motion, the nonlinear properties of muscle contraction, and the forces enacted by multi-joint muscles in more complex movements. After an increasing number of contradictions to the findings of the equilibrium point hypothesis [257,258], methodological criticisms [259] and slowly spreading biomechanical findings [260], Schmidt constrained his model to movements in which neither gravitational nor inertial forces were involved [261]. Thus, the VP model is not applicable to sports movements. Despite the rather narrow areas of corroboration, there is great merit in the fact that the VP model has inspired and encouraged more variable training in sports overall—even because of the different uses of the term in colloquial and scientific language. In addition, more attention was given to the question of how a movement can be learned more effectively. 

A similar development can be observed with the CI model. An analysis of the WM, which is the basis of the explanatory approaches, provides a more detailed explanation of the limitations of the CI model. The explanations refer to Baddeley and Hitch’s [165] WM model for short-term information processing with limited capacity. This model has been experimentally studied only for a capacity of up to seven cognitive items [262] and exclusively for visuospatial, sequential, and phonological stimuli [263]. This WM model has not been tested hitherto for tactile, kinesthetic, and proprioceptive aspects with multiple parallel scopes [263]. Supporting evidence for different pathways and brain areas with varying capacities for processing sensory-specific information is provided by more recent brain studies [264,265,266,267]. A further development of Baddeley’s [165,268] WM model is offered by the cognitive load theory [269,270], which considers to a greater extent the internal and external load on limited WM capacity. With the greater consideration of subjective information, ongoing access to long-term memory takes on greater importance and allows for the differentiation of novice and experienced participants independent of WM capacity [269]. In a broader sense, the idea was already included in the “concept of apperception” used by Wundt [271], whose ideas Peirce admired [272], to which also Uexküll refers in his differentiation of the biologically perceived environment into an objective and a subjective one [273]. In addition, Gestalt psychology takes the issue up in its principles of perception in the sense of a differentiated holistic view [274]. Aspects of it became popular in ecological psychology with Gibson’s affordances [275]. Against the background of a subjective environment, a separation of individual and environment in the context of sport pedagogical measures [23,93] seems scientifically outdated. 

In this context, a recently observed renaming of contextual conditions to constraints astonishes [24]. Although initially the introduction of a constraints-related approach was intended to extend the prescriptive teaching approaches [23] in the field of motor learning research, they were already identically described by the term ‘context’ (“Context is the interrelated characteristics conditions that exist in the individual, task, or environment at the time of the action” [276]). Because of ignoring the cautions in the use of the term ‘context’, it is little wonder that the constraints-led approach is meanwhile suffering the same fate as was formulated for context by Smith, Glenberg, and Bjork (p. 342): “Context … is a conceptual garbage can that denotes a great variety of intrinsic and extrinsic characteristics of the presentation or test on an item” [277]. Consequently, in analogy to Magills [276] statement about context, every event occurs under some constraints; the term constraint can be used so ubiquitously that it loses its force as an explanation. Epistemologically, both terms principally relate to the Hempel–Oppenheim Schema for scientific explanations where the explanation of a phenomenon consists in the proof that the phenomenon obeys the known general laws, which are to be applied to the special boundary conditions. Although the scheme was designed for deterministic laws, it also shows great utility for statistical inference [278] and illustrates the need for specifying laws in addition to constraints, contextual, or boundary conditions.

Interestingly, all studies of the CI model to date have been limited to fewer than seven items to be learned, and explanatory approaches have relied on a WM model that has rarely been tested for motor, tactile, and kinesthetic tasks [263]. Although the VP model became history after extensive inspiration, there are still many unanswered questions regarding the ideas of the CI model, despite extensive studies and claim limitations, especially in terms of the model’s sensory specificity. In addition, the concretization of randomness in the context of the law and the relative similarity of the tasks will be the subject of research. When determining the context in the form of several movement tasks, no statements have been made so far about their relative similarity or their maximum range of similarity. Depending on this, the changes in the context of a random sequence could bring different effects. For example, should random variation occur between a handball goal throw, soccer dribbling, and a volleyball serve, or should three different serve techniques in volleyball, or two volleyball serve techniques and a handball goal throw serve as the basic algebra?

### 4.3. Models Originating in Systems Dynamics and Biology

The third group is the DL models which can be considered one specific application of more general principles that originated in system dynamics and biology. With the introduction of subjective information using the stochastic resonance (SR) metaphor, the original model was extended. Compared to the use of SR to exclusively acutely improve balance control in the elderly by various kinds of mechanical stimulation of sensors on the soles of the feet [279,280], the use of SR within the DL model aims at stimulating a complex neural network and its sustainability [194,281].This expansion not only allowed other motor learning models to be described by means of the same underlying principle, but also built a bridge to Cybernetic Pedagogy through a stronger emphasis on the learner without neglecting the teacher’s role. In contrast to the models of the second group, here the model’s generalization trials did not take place by horizontal extension but instead by an additional vertical derivation from the underpinning principles of system dynamics and biology, namely the principle of self-organization by amplification of fluctuation and the Darwinian principle of variability.

More specifically—and in analogy to person-identification as the basis for the original DL model—after having identified the individuality of team behavior with AI methods [281,282], indications of the applicability of the DL model for tactic training were provided and corroborated [283,284,285]. The first extended applications of the DL model on movements with sDGF and in the field of rehabilitation showed positive corroboration as well [286,287,288,289,290,291]. A recent study indicated that the effects may be dependent on the complexity of the intended movement [292] with a tendency of the DL model to exhibit bigger effects on the more complex movements accompanied by small, nonsignificant disadvantages for less complex movements, which aligns with previous predictions [194].

In sum, the DL model has been developed out of an increasingly accelerating rethinking in the natural sciences shifting towards complex adaptive systems, combined with a fundamental paradigm shift in the philosophy of science. Both fields relinquished some degree of original control by accepting the idea of a comparably reduced predictability in large parts of their field of action, the one by Duhem and Quine’s underdetermination hypothesis and the other by the findings on irreversibility and self-organization. If the former was accompanied by the most extensive abandonment of belief in an absolute truth and led to the wider spreading of neo-pragmatism [228,293,294], the latter pointed to a core problem of the criterion of replication in scientific studies on living systems that is directly related to the major assumptions of the DL model. If a system is changing continuously, then the memorization of numerous repetitions in form of a single motor program is hardly justifiable. Although the impossibility of identical repeatability and the belief in Laplace’s demon [295] has been out of serious discussions at least since the three-body problem [296,297], the uncertainty theory [298], and the incompleteness theorem [299], this issue is of specific importance in the context of research on motor learning processes. While the problem of time as an influencing factor in group studies could still be circumvented to a certain extent by running simultaneous studies, the problems with the repetition of an experiment with the same group of persons are of a different dimension. That is, although the research subject would represent the greatest similarity for a replication study in life sciences with the same external context or objective information, the internal context or subjective information would have changed, nearly independent of the time elapsed between the two tests [229]. Biological memory in humans refers not only to neuronally related cognitive memory but also to the biology of tissues and metabolism [300]. Strength tests, as well as endurance tests, tend to leave traces all over the body, just as in motor learning processes, even if they are performed only once. Recent studies show that this difficulty is even worse than had been feared. With no other intervention but time alone, a person’s gait pattern appears to change by 85%–95% within a day [225] or between days [227]. Studies on musicians performing the same piece of music on the same and different days suggest similar results [301]. Because the replicability of studies in physics or chemistry could not be comparably achieved in most objects of interest in sciences that study humans and their behavior [302] statistics and probabilities were employed for “taming the chance” [303]. Although more or less stable and replicable traits and abilities, such as in Fitts law [81] have been identified within the scope of possibilities, constant changes—a fundamental characterizing property for living beings—have been neglected thus far, most likely due to a lack of methods or through elimination by detrending algorithms. Interestingly, the eliminated trends have rarely been objects of interest. Circadian rhythms [304,305,306] and phases of ontogenesis [307,308,309] have so far been defined largely independent of individuals and widely neglected in connection with motor learning.

## 5. Perspective

### 5.1. From Population to Group to Individual to Situation and Back

With the adoption of “statistical rituals” [18] from medical and psychological sciences, and due to the pressure for increased research output during the Cold War, the objects of investigation in life sciences became increasingly adapted to the methods available rather than vice versa. The same occurred in exercise and sports sciences, which were thus limited to only a few research questions [310,311]. Based on Gauss’s central limit theorem, the majority erroneously concluded that the replication demanded in science could be achieved only with large sample numbers. From a logical–statistical and pragmatic point of view, this assumption raises fundamental problems. The first is that the mean values of a sample can only be inferred to the mean values of another sample within the same population, but not to other populations or a single subject [312]. This crucial but widely unknown fact (i.e., that the mathematical theory of probabilities cannot be applied to factual probabilistic situations) was described by Kolmogorov and is denoted the application problem of probability [313,314]. In addition, fully defining the similarity of samples and populations encounters the problem of Duhem and Quine’s holism thesis. This problem is often in parallel closely related to the mixing of the meaning of the term “true” in logic with that in the statistics-based sciences. While in logic it is—along with “false”—a value for the validity of statements or conclusions, in statistical sciences it represents a psychological concept referring to a population where the mean value is defined as the estimated “true” value [315]. From a pragmatic point of view, knowledge about individual behavior is necessary to train top athletes or to effectively treat patients during rehabilitation. Reducing top athletes to a general prototype of a respective discipline rarely leads to Olympic medals, and treating patients based solely on the main symptoms of an illness rarely leads to complete convalescence; both neglect the long-known and accepted individuality of world-class athletes and patients. 

In this context, group and single-case studies are, structurally, only two extremes of development that can already be observed in group studies [312]. Growing difficulties in achieving replicability or predictability in non-specific group studies increasingly led to studies with more specific populations. Studies are conducted separately by sex, by age, by performance level, by handedness, by culture, by sport, by time of day or year, etc. Structurally, these specifications are intermediate steps in the framework of advancement beyond the specificity of the individual to the greatest specificity, i.e., the situations of the individual. 

Even though individuality has been constituent of modern sports and movement science from the very beginning by the training principle of individuality [75], it was hitherto realized in the practice of elite sports and the accompanying medical–biomechanical performance diagnostics only in the form of individual metabolic and coordination profiles [94,316]. In exercise and movement science, after the banishment of individual case studies from psychological research in the postwar years, with a few exceptions (primarily related to personality research [317,318]), it mainly paid lip service to individuality of movement or movement learning and was preferentially applied in the form of individual case descriptions [319] as the normative counterpart of group analyses using the same frequency statistics [320,321]. Similarly, the use of the term “individual” is frequently found in conjunction with a normative demand for individuality in the absence of group assignment [322,323,324,325]. Visual inspection of several individual learning curves was long used but can at best provide only the first coarse indications for individual learning behavior without meeting the basic forensic requirements of uniqueness and persistence. In addition, longitudinal studies are primarily based on discrete results such as jump height [326], balance scores [327], or hitting performances [328]. Biomechanically based longitudinal studies on movement data are frequently either limited to time-discrete movement characteristics [212] or to describing average time-courses with a standard deviation [329,330]. Suggestions for a theoretical decomposition of the variance of movements were provided with the uncontrolled manifold [331] and the three-component method [332] but have hitherto been restricted to movements with sDGF and target-oriented movements. However, the role of noise in these approaches neglects the successful assignment to situations with varying emotions, fatigue, or music that the subjects were listening to [222,223,224]. All these situations lead to a finer resolution of the previous noise distribution around the mean. To what extent potential remaining cross-joint reflexes during movements with lDGF may be considered in the two approaches requires additional research. Methodologically, in most approaches, individuality is assumed as a given without having been examined in terms of the two forensic criteria. Accordingly, individuality is assumed only if the criteria of uniqueness and persistence are satisfied [333]. Although this could be shown extensively in the field of single movements on different levels of observation by means of nonlinear pattern recognition [334,335,336], this represents a particular challenge for the proof of individuality of learning behavior due to persistence. Without explicitly referring to the replication crisis, a series of studies by Horst et al. [225,226,227] on the reliability of gait patterns over different time scales takes up precisely this aspect of natural change of the research object over time and proposes a way to address it.

### 5.2. From Time-Discrete Frequency Statistics and Model-Oriented Prescriptions to Process-Oriented Bayes Statistics and Self-Organization

Five approaches suggest coping with several of the problems related to the analysis of movement learning as a temporally continuously changing interaction process between coach and athlete, or between therapist and patient, with a stronger focus on the interpretant of the content to be learned. More sensitive diagnostic tools for process-oriented movement analysis are combined with pattern recognition methods based on AI that has been developed in parallel with the advancement of the DL model [211,219,229,337].

First, instead of supporting the analysis exclusively with error-sensitive time-discrete data, a process-oriented analysis in combination with pattern recognition not only enables an automatic and parallel differentiation of movement techniques (e.g., walking, running, or jumping) but also modes of movements (e.g., happy, sad, tired, or limping) or movement styles (e.g., individual walking styles), as they are essential for the quantitative analysis of learning and therapy processes when qualitative changes occur [126,338]. Interestingly, the problem of the similarity and complexity of exercises with higher parallel processing demands in the context of the CI model could be approached quantitatively [339]. 

Second, all these analytic differentiations rely mainly on different levels or structures of noise that could be used practically for interventions. Applying the adequate amount and structure of noise—which is dependent on the situational context of the individual—would increase the spectrum in therapy and training enormously, since it would transform from group-specific to individual- and situation-specific. Irrespective of differentiation, the DL model seems to provide advantages in teaching groups compared to only a few methodical exercise steps [230].

Third, recent developments in the analysis of large sets of movement data by means of deep learning applications provide a further alternative approach for decomposing objective and subjective information by layer-specific information content [340]. This is of specific interest for investigating the individuality of learning in terms of persistence. Because learning the same movement a second time is influenced by the first time, the learning of different movements has been suggested as an avenue to find individual learning characteristics [229]. Across different disciplines, individual common characteristics may be identified that, in the next step, could be used to make therapy and training more effective. Fluctuations of the movement signal in the context of a motor learning process are decomposed into person-independent signals, which are assigned to a movement component or external objective information and an individual-specific signal component as internal subjective information. In parallel, individual physiological signals, such as EEG-frequency spectra or heart rate variability, could be taken as indicators of subjective information [341]. In this context, these signals can be additionally decomposed into emotion- or fatigue-related modifications interpreted as situated individual information. A further challenge would be to combine this approach with analysis based on frequency content [342]. 

Fourth, if the analysis of changes in movement patterns is consistently taken to changes without targeted intervention, the sensitivity of AI methods now allows the discrimination of an individual’s gait patterns within a day [226], from day to day [225], and within a year [227], with detection rates of 85%, 95%, and 99%, respectively. These analyses point to the above-mentioned characteristic of living systems that change constantly and irreversibly. With a more differentiated analysis of the noise changes on different time scales [328], individual phases of stability could be identified via the rates of change, which support interventions. The extent to which the apparently unmotivated changes are due to genetic aging processes [343,344] or to variations in the Earth’s magnetic field [345,346,347] opens up extensive further areas of research, again supporting Duhem and Quine’s holism argument and the problem of absolute knowledge. If temporal change is a characteristic of the object of study, then it should be studied in more detail in the movement and, especially, learning sciences, in analogy to the properties of a rigid body in physics. Neglecting this aspect will continue to keep the uncertainty of studies on living systems high and the rate of replicability low.

On the one hand, obtaining more information about the individual in specific situations would enormously expand the knowledge of subjective information on the learner’s side, depending on objective information in the sense of Peirce’s interpretant. A promising approach in this direction is already offered by the action type system [348]. On the other hand, the nonlinear interactions of all possible influencing variables and the limited measurement accuracy will always limit deterministic predictions, keep the research process active, and continue teaching us to accept uncertainty. 

Fifth, AI-based pattern classification offers an approach to bridge the two extremes of group and single case studies by means of a continuous shift of the level of interest within the same analysis. Current developments in data acquisition using wearable sensors and image recognition provide the corresponding basis, at least for collecting the necessary amount of movement data. Support in the same direction is provided by the close relationship of pattern recognition to Bayes statistics with a different epistemological basis from the classical frequency statistics [349,350]. This approach also supports a research strategy in the political dimension that is less centralized, less focused on selected programs, and less susceptible to specific politics. The distributed recording of longitudinal studies on single cases and their anonymous saving in data banks [351] in each laboratory could be requested for specific research questions to achieve the sample sizes necessary to allow classical statistics. Having a distributed collection of data from different cultures would enormously broaden the understanding of our research object with its temporal changes, and could lead to a scientific field without physics envy for replicability [9,12].

## 6. Conclusions

A coarse historical and epistemological analysis of the movement learning models applied and discussed in schools, clubs, and clinics reveals a process of knowledge as described by post-analytical philosophy. With each model, historically, new aspects of an infinite spectrum of influence come into play, which are rarely compared with each other [352] and exist in parallel [353]. In each case, all models fulfilled their model purpose, which is best described by their beginnings and first experiments. However, the historical development of most of the models points to increasing specificity, and thus a departure from the initial generalization claims. Nevertheless, all models were—and remain—inspirations for a wide environment beyond their community. Instead of understanding the models as scientifically proven theories and marketing them to the public as absolute facts, from the perspective of the theory of science, one would always have to assume only provisional proof. Fisher expressed himself in similarly cautious terms in his statistics when he said that a 5% level merely indicates that further research is worthwhile [354]. Historically, the introduction of the criterion of replication as an essential component of scientific work is always in close connection with the machines on which mankind had to rely for security, to secure survival, and to use available capacities otherwise. A car that behaves like a pubescent teenager would probably not find a large sales market. A prerequisite for this phenomenon, however, was the detailed knowledge of the properties of the object of research, which are historically quite extensive in mechanics, electromagnetism, optics, etc., and have the merit of being stable on the time scales in which we observe them. Nonetheless, the first human-developed machines, beyond a certain level of complexity, do not always exhibit predictable behavior. In comparison, the constant changes as a characteristic property of the object of research in movement and sports science are so far only rudimentarily studied and understood. Closely related to this is the neglect of the study of the individual [193]. Both require the application of other methods in both data analysis and statistics that have been increasingly developed for this purpose. It is up to scientists and practitioners to take advantage of the knowledge of previous generations, not only to consider the ‘what’ and ‘how’ of a learning content but also the ‘why’, and initiate changes rather than renaming inventions that were new some time ago.

## Figures and Tables

**Figure 1 ijerph-19-00711-f001:**
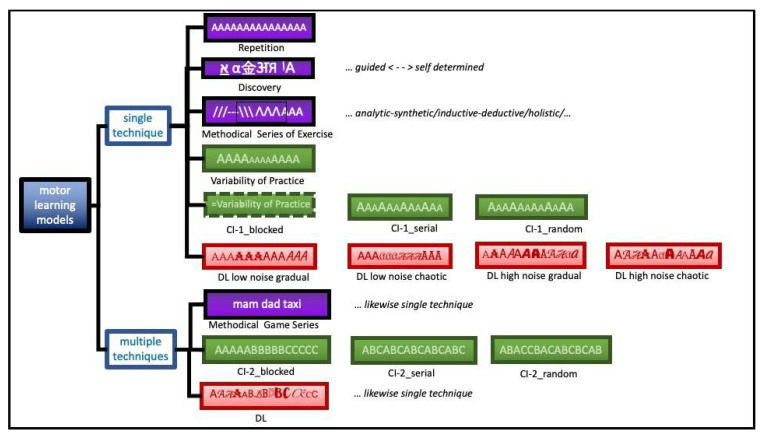
schematic visualization of the motor learning models discussed. A, B, and C correspond to different movement techniques. The three colors correspond to the origin of the models. Purple: pedagogical origin; green: sport psychological origin; red: system dynamic origin. CI: Contextual Interference Learning; DL: Differential Learning.

**Figure 2 ijerph-19-00711-f002:**
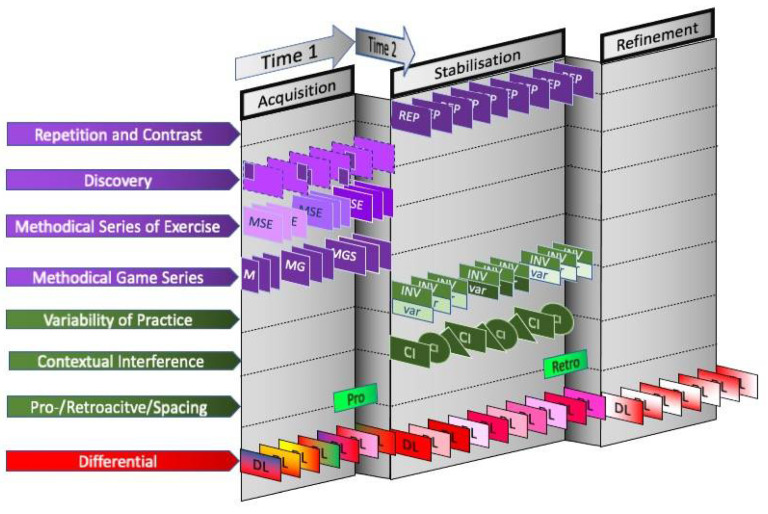
The motor learning models structured by their time-scale-dependent area of application according to their original model claims.

**Table 1 ijerph-19-00711-t001:** Summary of the motor learning models including the original model claims and objectives of research.

What?	for Whom?	When?	for What?	Guidance		Degrees of Freedom		Processing		Main Sensory System		
				External	Self-Organized	Small	Large	Sequential	Parallel	Visual	Acoustic	Kinesthetic
RL	teachers and educators	500 BC	learning in general	x		x		x	x	?	?	?
DBL	teachers and educators	500BC and 1762	individualised learning		x	x	x		x	x	x	x
MSE	sports pedagogues and physical education teacher	1820/1930	acquisition of complex movements, ind. starting level	x			x		x			x
MGS	sports pedagogues and physical education teacher	1960	acquisition of team games, ind. starting level	x			x		x	x		x
VP	movement scientists with sport psychological background	1975	stabilizing ballistic, automatized movements	x		x		x		x		
CI	psychologists with a movement science background	1979	stabilizing movements	x		x		x		x		
DL	high performance coaches/biomechanists/motor control scientists	1999	refining and stabilizing high performance techniques	x	x		x		x	x	x	x

RL: Repetitive Learning; DBL: Discovery-based Learning; MSE: Methodical series of exercise; MGS: Methodical game series; VP: Variability of Practice Learning; CI: Contextual Interference Learning; DL: Differential Learning.

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
