# Peer review of "Always Pay Attention to Which Model of Motor Learning You Are Using"

_ijerph, 2022, doi:10.3390/ijerph19020711_

Round 1
Reviewer 1 Report
Dear Authors This is one of the most interesting and important texts for the field of Motor Learning that I have had the opportunity to read recently. The manuscript appears to come from a German-language thesis. As a researcher in the field, whose language is non-Germanic - I certainly want this text published as soon as possible for academic purposes. But as a scientific reviewer, I realize that there are some important questions, which I will now raise: 1. The text is dense and complex, which I think is great. However this has to be in line with clear and objective language. I know it is quite difficult to bring what has been covered in depth in a thesis for the appreciation of readers who will not always be familiar with certain concepts and theories. But some effort must be made so that readers can follow the author's thoughts. A complete review of the text, seeking to remove any information that did not contribute to the achievement of specific objectives would be very desirable. 2. The Introduction, despite offering a suitable backgroud, seems to need some improving. (a) make it clear to the reader which theoretical approach is chosen, avoiding doing this when the objectives are stated; (b) redo the objectives in order to make them more concise and precise;3. Every scientific paper has a Methods section. And this one is no exception. If this is a review study, describe what type of review it is. Be clear and concise. Make clear the delimitation of the review - either theoretical-based or time-based.
4. Make sure that all the main topics developed in the text correspond to each of the objectives proposed in the review.
Author Response
Please also open the attached file
Response to Reviewer 1
This is one of the most interesting and important texts for the field of Motor Learning that I have had the opportunity to read recently. The manuscript appears to come from a German-language thesis. As a researcher in the field, whose language is non-Germanic - I certainly want this text published as soon as possible for academic purposes. But as a scientific reviewer, I realize that there are some important questions, which I will now raise:
We would like to thank the reviewer for your thoughtful comments on manuscript “Always pay attention to which motor learning model you are using. Please see our responses to your comments:
- The text is dense and complex, which I think is great. However, this has to be in line with clear and objective language. I know it is quite difficult to bring what has been covered in depth in a thesis for the appreciation of readers who will not always be familiar with certain concepts and theories. But some effort must be made so that readers can follow the author's thoughts. A complete review of the text, seeking to remove any information that did not contribute to the achievement of specific objectives would be very desirable.
Response: Thank you, we put an effort to address these concerns – the language gained more objection as citations were provided (i.e., The term "Envy of physics" relates to the envy by scientists in other disciplines of the mathematical precision of fundamental concepts achieved by physicists. It is a criticism leveled against academic areas, such as social and life sciences, that attempt to express their fundamental concepts in terms of mathematics, which is seen as an unwarranted push toward reductionism. Nelson, R. R. Physics Envy: Get Over It. Issues Sci. Technol. 2015, 31 (3), 71–78.. In addition to the suggestions above, we intended to remove the non-specific information (i.e., Differential learning model (DL) Description and Historical context). However, we think that the recommendation to remove "any information that did not contribute to the achievement" addresses a key point raised by reviewer 2, namely that it can be often difficult to establish a link to the specific objectives because the aspects are not sufficiently explained. Due to the density and complexity of the information, we have therefore decided to include a few supplements for comprehensibility in line with reviewer 2's recommendations. In addition, we believe that shortening other parts would not facilitate the reading, it would further require specific preparation (concepts and theories) for this article.
- The Introduction, despite offering a suitable backgroud, seems to need some improving. (a) make it clear to the reader which theoretical approach is chosen, avoiding doing this when the objectives are stated; (b) redo the objectives in order to make them more concise and precise;
Response: the chosen theoretical approach (critical review) is described in 3, the objectives are formulated more precisely and the explanations of the theoretical approaches within the objectives are removed (line 139-173).
1) Introducing two areas of research whose research subject matter is closely related to the implications of Peirce's "interpretant" for theories of motor learning and their implementation. Both provide the criteria for a template against which selected motor learning theories are discussed.
- a) Deriving criteria from the GeMoT for a clearer restriction of the previous generalizations of motor learning models related to the researcher as “interpretant”.
- b) Deriving criteria from Cybernetic Pedagogy for differentiating objective and subjective information in the context of a motor learning process related to the learner as “interpretant”.
2) Illustrating and discussing the implications of the importance of these two areas on the most common motor learning and teaching approaches that
- a) are found in textbooks on physical education, the training of athletes, physical therapy, and occupational therapy,
- b) introduced new elements related to the physical exercise process, and
- c) are subjects of scientific research.
3) To suggest an alternative approach to problems particularly related to the replication crisis through recent developments in the recognition of motion patterns using artificial intelligence. This approach offers an alternative in dealing quantitatively with the object of research of moving and learning humans in the form of locally generalizable statements related to the non-repeatability of events, which are commonly and somewhat succinctly attributed to the factor of time.
- Every scientific paper has a Methods section. And this one is no exception. If this is a review study, describe what type of review it is. Be clear and concise. Make clear the delimitation of the review - either theoretical-based or time-based.
Response: the whole paragraph has been restructured and rephrased (line 180-188).
The objective of this study is a critical review [45; Grant et al 2009] on the most common motor learning approaches based on criteria that have only sporadically been considered in the previous research. These criteria are derived from two theories that can be traced back to the theory of semiotics according to Peirce [43], namely the General Model Theory (GeMoT) and the Cybernetic Pedagogy .
Due to the breadth of the topic and limitation of space and time, the individual areas can only be touched upon. Naturally, the older motor learning theories have a more extensive historical context, whereas with increasing topicality the scientific foundation widens in scope. This work makes no claim to completeness.
- Make sure that all the main topics developed in the text correspond to each of the objectives proposed in the review.
Response: We feel that the main topics developed in the text correspond to each of the objectives proposed in the review. If reviewer has specific concerns, we would like them to be notified more precisely.

Reviewer 2 Report
First of all I want to congratulate the authors of this manuscript for the very interesting, thorough, and extensive presentation of the most important theoretical approaches regarding motor learning and control. I read the work with pleasure and found it well written and comprehensive.
Given the complexity of the topics covered, it is likely that various aspects or specific terminologies of the theoretical approaches presented are in some ways not completely clear to non-expert readers. My comments, therefore, are limited to requests for minor clarifications or elaborations to make the work fully understandable to a wider audience of readers.
1) L 50-52. "…physics envy…") not clear.
2) L 53. Do you mean replication studies or replication studies finding similar results?
3) L 58. What do you mean with "collecting constraints"?
4) L 59-60. I think this sentence should be made clear for non-expert readers.
5) L 71-72. Unclear, please elaborate.
6) L 100. Can you provide examples of repetition learning and interleaved learning?
7) L 176. What kind of differences?
8) L 184. What is intended with “interpreted generously”?
9) L 204-205. Why redundancy for the learner reduced allows learning to occur?
10) L 176. What kind of differences?
11) L 186. “a process of gaining redundancy”, unclear.
12) L 223-224. “intention-ho-o mogeneous group”, unclear.
13) L 271. What kind of external stimuli?
14) L 280. What other goals?
15) L 316. Should be "Figure 1" not "Figure 1a"
16) L 316-318, Figure caption, please include and spell out "Meth. Ser.", "Var.Pr." "dto", “DL”.
17) L 378. “naturalistic fallacy”, unclear.
18) L 381. In which way RL is similar to the constraint led approach? It seems to me that they are largely different approaches.
19) L 396-397. Unclear, please elaborate.
20) L 427-428. What is intended with "conditionally appropriate and unconditionally appropriate"?
21) L 445. What is intended with “quasi-executive functions”?
22) L 474-475. “corresponds to the consideration of subjective information”. Unclear.
23) L 547. Isn't it quantitative normative?
24) L 645. This study was conducted in badmington, not in baseball.
25) L 657-658. This sentence is difficult to understand, please elaborate.
26) L 663. What are the two phenomena?
27) L 681. Please define stochastic perturbations.
28) L 721-722. “situational dependence on…” and also individual differences?
29) L 737-738. “both types of learning”, do you mean assimilation and accomodation?
30) L 766. “4.8 Graphical and tabular summary of the results”. Shouldn't this heading be in bold format and not be in italics format?
31) L 768. Figure 1a is not mentioned in the text, thus “figure 1b” should be Figure 2.
32) L 768. If "results" refers to both table and figure should be "result".
33) L 772, “figure 1b”. See previous comment.
34) Table 1. Excellent summary, congratulation!
35) L 780. Figure 1b should be Figure 2. Excellent representation!
36) L 830. “transform multiple goals into abstract models”. Unclear.
37) L 890. Shouldn't be "Magill's"? Also add a citation.
38) L 904-906. Sentence unclear.
39) L 939-942. Difficult to understand.
40) L 977. “values of a sample only be inferred only to the mean values” Is something wrong/missing in this sentence?
Author Response
Please also open the attached file
Response to Reviewer 2
First of all I want to congratulate the authors of this manuscript for the very interesting, thorough, and extensive presentation of the most important theoretical approaches regarding motor learning and control. I read the work with pleasure and found it well written and comprehensive.
We would like to thank the reviewer for efforts towards improving our manuscript “Always pay attention to which motor learning model you are using“. In the following, we address your comments
Given the complexity of the topics covered, it is likely that various aspects or specific terminologies of the theoretical approaches presented are in some ways not completely clear to non-expert readers. My comments, therefore, are limited to requests for minor clarifications or elaborations to make the work fully understandable to a wider audience of readers.
- L 50-52. "…physics envy…") not clear.
Response: Thank you for the comment, the following explanation is added (line 52-56):
The term "Envy of physics" relates to the envy by scientists in other disciplines of the mathematical precision of fundamental concepts achieved by physicists. It is an criticism leveled against academic areas, such as social and life sciences, that attempt to express their fundamental concepts in terms of mathematics, which is seen as an unwarranted push toward reductionism. (Nelson, R. R. Physics Envy: Get Over It. Issues Sci. Technol. 2015, 31 (3), 71–78) - L 53. Do you mean replication studies or replication studies finding similar results?
Response: We apologize, but we do not exactly know what is meant here. There is a misunderstanding as we wrote about replication rates (not studies) which describe the results of two replication studies quantitatively in relation to each other. - L 58. What do you mean with "collecting constraints"?
Response:
… propose to reintroduce long-lost belief in Laplace’s demon by collecting all the constraints that have been used in intervention studies and are intended to make motor learning predictable… - L 59-60. I think this sentence should be made clear for non-expert readers.
Response: Edited (line 58-70)
However, whereas in the other life sciences, various causes for the crisis have been discussed and alternatives have been proposed that have the potential to bring about change and progress, most sports science publications prefer to persevere with traditions. The majority of sports science publications still “blindly follow the ritual of mindless statistics” [16–20]; or propose to reintroduce long-lost faith in the Laplace demon that relies on the belief in predicting the future by knowing the past, by collecting as many boundary conditions as possible, that have been and are being used in intervention studies to make motor learning predictive [21–24] or seek to renew the belief in absolute falsification according to Popper’s theory of absolute truth [25,26] despite the fact that the corresponding positivism and neo-positivism was overcome already in the middle of the 20th century and lead to post-analytic philosophy. - L 71-72. Unclear, please elaborate.
Response: We added the following explanation (line 82-84):
This approach is advocated primarily by researchers in the life sciences whose objects of study are subject to cultural differences or constant change, such as on learning, ontogenetic, or phylogenetic time scales. - L 100. Can you provide examples of repetition learning and interleaved learning?
Thus, Newton’s law of gravity developed on rigid bodies is no more disproved by a falling down than repetition learning is disproved by interleaved learning.
Response: This is an excellent question.
Similarly, to the first case, the results in the second case are tied to the properties of the object of study and the conditions of study. If in one case it is the ratio of the surface to the weight and the surrounding properties of the medium, in the second case it is the properties of the living being such as age, learning experiences, degree of openness to new things, need for security, complexity of the movement to be learned on which the result will depend. - L 176. What kind of differences?
Response: What we had in mind is that in pedagogical cybernetic theory, it is assumed that the learner will gain mastery over their behavior in specific contexts by detecting differences … between the earlier and the current perceptions and experiences - L 184. What is intended with “interpreted generously”?
Response: Thank you, changed, see line 241-244). When interpreted generously, since it was never explicitly associated with this cybernetic view, subjective information was at best accounted for by differentiating findings between beginner and advanced learners, children and adults, or specialists in different areas. - L 204-205. Why redundancy for the learner reduced allows learning to occur?
Response: In consequence, only when repetitions are not identical and show differences, which Bernstein [61] metaphorically described as repetition without repetition, where redundancy is reduced for the learner while increasing subjective information as their complimentary so that learning can occur. - L 176. What kind of differences?
Response: This question is in line with the question 7. - L 186. “a process of gaining redundancy”, unclear.
Response: This expression cannot be found in line 186 but is in 266 (in the revised version). There it mainly refers to the same as in question 9. Nonetheless, we extended the explanation to ensure more fluidity of the text for a better comprehension of the reader.
Learning thus acquires a strongly subjective component and can be understood as a process of gaining redundancy that depends on the information already available and corresponds to a decrease in subjective information - L 223-224. “intention-ho-o mogeneous group”, unclear.
Response: The sentence has been rephrased (line 288-290):
This is an open system insofar as the intentions, purposes, and goals are decided within the framework of modeling in the context of a group of people with sufficiently similar intentions for a certain time, related to the model criteria. - L 271. What kind of external stimuli?
Response: The following supplements have been made (line 336-338)
… with the after-effects of behaviorism, increasingly restricted itself to the optimization of measurable behavior through visual, auditory etc. stimuli
L 280. What other goals?
Response: Thank you for your question. In this context the other goals refers to the goals that go beyond purely motor goals that are most often associated with psychology, sociology, nutrition, life style, injury prevention etc. .
- L 316. Should be "Figure 1" not "Figure 1a"
Response: Adapted - L 316-318, Figure caption, please include and spell out "Meth. Ser.", "Var.Pr." "dto", “DL”.
Response: Adapted - L 378. “naturalistic fallacy”, unclear. an informal logical fallacy which argues that if something is 'natural' it must be good.
Response: The citation for the better comprehension of the text added (line 449): Moore, G. E. Principia Ethica; Cambridge University Press: Cambridge, UK, 1903. - L 381. In which way RL is similar to the constraint led approach? It seems to me that they are largely different approaches.
Response: Rephrased (line 451-458):
is currently experiencing a renaissance in competitive sport with a new name, the con-straints led approach (CLA), where material or instructional aids are also used to con-strain the learner's movements in a way that guides him or her in the direction of the coach’s learning aim [92,93]. However, while in RL errors are to be avoided and indi-vidual techniques are tolerated only at the highest level of performance as an inevita-ble by-product [75,94 ], in CLA erroneous movements are allowed but only to experience how not to do them and individual techniques are accepted but only within the guiding constraints [93]
- L 396-397. Unclear, please elaborate.
Response: Thank you, this sentence is rephrased to read (line 474-478): Proponents of the RL approach still claim general applicability regardless of learning levels, age, or complexity of a movement. This is experiencing continuation in cognition-oriented feedback learning by means of outcome or performance knowledge, where the notion of correct repetitions is mainly accompanied only by cognitive components [56].
- L 427-428. What is intended with "conditionally appropriate and unconditionally appropriate"?
Response: an explanation for inappropriate is included (line 509-510):
… inappropriate, in the sense of not suitable for teaching, … - L 445. What is intended with “quasi-executive functions”?
Response: this part of the sentence was rephrased (line 527-529):
Stabilizing and perfecting skills is expected through the parallel acquisition of abilities that today would be called executive functions, which are developed primarily through self-activity and self-motivation - L 474-475. “corresponds to the consideration of subjective information”. Unclear.
Response: The sentence is rephrased (line 556-559):
“From the point of view of cybernetic pedagogy, adjusting the difficulty level of the exercises to the learner’s level of knowledge at the beginning corresponds to taking subjective information into account” - L 547. Isn’t it quantitative normative?
Response: The sentence was rephrased (line 633-635):
The first publications were mainly qualitative-normative, which were later accompanied by quantitative descriptions of the realization a finer scale for evaluating the given norms. - L 645. This study was conducted in badmington, not in baseball.
Response: edited - L 657-658. This sentence is difficult to understand, please elaborate.
Response: The sentence is rephrased and the explanation is extended (line 744-750): This in turn corresponds to the general principle of cybernetic pedagogy, according to which the subjective information to be assimilated per unit of time represents an individual constant. The subjective information absorbed in a first learning step represents an evolved redundancy for the learner. In order to achieve the most effective learning process for the individual, the amount of information for the next task would have to be increased by exactly this amount in a further learning step, i.e. it would have to contain something new. - L 663. What are the two phenomena?
Response: The sentence was rephrased to (line 758-760):
the two phenomena of intra-task interference and inter-task facilitation associated with the CI model inspired the entire research field with the standardized distinction of acquisition and learning phases in most motor learning studies [191] - L 681. Please define stochastic perturbations.
Response: Thank you. We have added the following (line 829-833):
To keep the summary for all motor learning in comparable length we gave the following explanation later in the text. A stochastic perturbation is understood here as any random sequence of variations in elements of movement, where the perturbations may have an internal or external origin. In this context, the term random is interpreted in a mathematical sense, which also includes repetitions of numbers as well as strictly monotonic or cyclic sequences that are not considered as such in everyday life [197]
28) L 721-722. “situational dependence on…” and also individual differences?
Response: We have added individual for more clarity (line 818-819)
“… their individual and situational dependence on “
29) L 737-738. “both types of learning”, do you mean assimilation and accomodation?
Response:
Most intriguingly, both types of learning, assimilation and accommodation or stabilization and acquisition/refinement, were achieved without having executed a single presumably correct movement and therefore massively contradicted all previous repetition-based models.
30) L 766. “4.8 Graphical and tabular summary of the results”. Shouldn't this heading be in bold format and not be in italics format?
Response: Thank you - adapted
31) L 768. Figure 1a is not mentioned in the text, thus “figure 1b” should be Figure 2.
Response: Adapted
32) L 768. If "results" refers to both table and figure should be "result".
Response: Thank you - adapted
33) L 772, “figure 1b”. See previous comment.
Response: Thank you - edited
34) Table 1. Excellent summary, congratulation!
Response: Thank you very much -highly appreciated
35) L 780. Figure 1b should be Figure 2. Excellent representation!
Response: Thank you very much - highly appreciated
36) L 830. “transform multiple goals into abstract models”. Unclear.
Response: the sentence has been rephrased to read as (line 933-936):
… a future challenge will be to translate the always multidimensional goals in pedagogy, such as affective, social, political, cognitive, and motor, into such abstract models that additionally depend on various individual and constantly changing conditions.
lines 920-923
37) L 890. Shouldn't be "Magill's"? Also add a citation.
Response: Thank you, reference 276 (line 998) has been added.
38) L 904-906. Sentence unclear.
Response: The sentence has been rephrased and extended by an explanation (line 1012-1020).
In addition, the concretization of randomness in the context of the law and the relative similarity of the tasks will be the subject of research. When determining the context in the form of several movement tasks, no statements have been made so far about their relative similarity or their maximum range of similarity. Depending on this, the changes in the context of a random sequence could bring different effects. For example, should random variation occur between handball goal throw, soccer dribbling, and volleyball serve, or should three different serve techniques in volleyball, or two volleyball serve techniques and a handball goal throw serve as the basic algebra?
39) L 939-942. Difficult to understand.
Response: This could be difficult for readers that are not aware of these concepts. Expecting this, we placed the corresponding reference directly beside the terms to give the possibility for further studies. We think that an explanation of all these concepts would go beyond the scope of this contribution. Therefore, we would like to stay with the present form.
40) L 977. “values of a sample only be inferred only to the mean values” Is something wrong/missing in this sentence?
Response: The sentence was rephrased (line 1095-1097):
The first is that the mean values of a sample can only be inferred to the mean values of another sample within the same population, but not to other populations or a single subject [312].
